# A General Framework for Empowering Graph Neural Networks with Causal Invariance

## Abstract

Graph Neural Networks (GNNs), despite their success, are fundamentally limited to learning a correlational mapping. We theoretically demonstrate that this limitation is inherent to the neighborhood aggregation paradigm of GNNs. This inability to distinguish true causality from spurious shortcut patterns leads to poor generalization ability. To bridge this gap, we introduce the Principle of Causal Alignment, a novel learning paradigm for GNNs, designed to empower GNNs with causal invariance without altering their architectures or compromising inference efficiency. We then present `CausGNN`, an instantiation of this principle. It employs a teacher-student strategy where a teacher GNN learns to compute the interventional distribution via backdoor adjustment, and then distills this causal logic into the student GNN, compelling it to learn invariant representations. Extensive experiments show that `CausGNN` not only improves the performance of various classic GNNs on node-level tasks but also exhibits superior robustness against noise and Out-Of-Distribution (OOD) challenges. The source code is available at: `https://anonymous.4open.science/r/CausGNN/`.

## 1 Introduction

Graph Neural Networks (Gori et al., 2005; Scarselli et al., 2008) have become a dominant paradigm for machine learning on graph-structured data (Duvenaud et al., 2015; Bronstein et al., 2017; Monti et al., 2017). Their power stems from the neighborhood aggregation (or message-passing) scheme, where nodes iteratively update their representations by aggregating information from their local neighbors. This fundamental mechanism allows GNNs to learn powerful representations of complex relational data, leading to outstanding performance across a wide array of applications (Tang et al., 2009; Wu et al., 2019; Yang et al., 2023).

Within this paradigm, a key evolution involves refining aggregation strategies (Kipf, 2016; Hamilton et al., 2017; Veličković et al., 2017; Brody et al., 2021). This progression ranges from early models like GCN (Kipf, 2016), which uses fixed, structure-based weights, to more powerful approaches. For instance, GraphSAGE (Hamilton et al., 2017) generalizes aggregation with various pooling functions (e.g., mean, max, or LSTM ((Hochreiter & Schmidhuber, 1997), while the Graph Attention Network (GAT) (Veličković et al., 2017) introduces an attention mechanism to weigh neighbors. However, the attention rankings in GAT are considered "static" as they are independent of the querying node. GATv2 (Brody et al., 2021) addresses this specific limitation by modifying the order of internal operations in GAT, creating a truly dynamic and more expressive mechanism.

Despite their great success, we argue that standard GNNs built upon the neighborhood aggregation paradigm share a fundamental limitation: *from a probabilistic perspective, the learning process of a GNN approximates the observational conditional probability $P(Y|X)$*, which means that it merely captures statistical correlations, regardless of the underlying causal structure. From a mechanistic perspective, the neighborhood aggregation process effectively functions as a sensor, perceiving and encoding a node's immediate local environment. Since these local environments are often rich with statistically prominent patterns, the model readily learns to exploit them to fulfill its optimization objective.

Let us consider a practical example from *ogbn-products*(Hu et al., 2020), a real-world dataset representing the co-purchase network of products on Amazon. The task is to classify a "battery" product.

Its intrinsic causal features, derived from its title and description (e.g., "Alkaline", "1.5V"), unambiguously place it in the "Accessories" category. However, they are frequently co-purchased with items like "electric toys" and "electronic watches" (i.e., its neighbors). A standard GNN is easily swayed by this neighborhood context, incorrectly predicting the batteries as "Electrics" category. We refer to this behavior of exploiting the environment for prediction as an environmental shortcut, and the dominant contextual information is termed shortcut features.

This reliance on environmental shortcuts is a general vulnerability present across various task settings. On in-distribution data, as the example shows, it leads to incorrect predictions in cases of contextual mismatch. Even in aligned cases where the shortcut appears to work, the learned patterns are inherently fragile. For instance, in noisy scenarios (e.g., with erroneous edges), the integrity of the environmental information is directly compromised. This vulnerability is amplified in Out-Of-Distribution (OOD) scenarios, as the statistical relevance of previously learned shortcut features becomes obsolete on the new distribution.

To address this limitation, inspired by invariant learning (Arjovsky et al., 2019; Krueger et al., 2021; Chang et al., 2020), we argue that a robust GNN requires learning patterns that remain stable across different environments. This will encourage the model to learn the true mapping between intrinsic node features and labels, ensuring it is not susceptible to shortcut features. In other words, we aim for the model to focus more on uncovering causal relationships (Pearl, 2009; Bühlmann, 2020) rather than merely relying on statistical correlations.

To achieve this goal, we propose a novel learning principle for GNNs, i.e., the Principle of Causal Alignment. This principle posits that a GNN can be guided to learn invariant causal relationships by forcing its predictions to align with a constructed, causally-debiased distribution. Building upon this, we then introduce `CausGNN`, a general teacher-student framework that provides an effective instantiation of this principle. The core idea of `CausGNN` is to learn cross-environment invariant representations using causal reasoning tools. Specifically, we exploit the *do-calculus* on the teacher model by constructing intervention pairs based on a backdoor adjustment criterion. It encourages the teacher model to learn invariant relationships between causal patterns and predictions by learning to approximate $P(Y|do(X))$, regardless of changes in the shortcut feature distribution. Subsequently, the teacher's learned causal logits are distilled into a student GNN. This process acts as a causal regularizer, compelling the student to acquire environmentally invariant representations without altering its architecture. During inference, the student GNN is employed standalone, which allows our framework to be seamlessly applied as a "plug-and-play" enhancement without sacrificing inference efficiency. Extensive experiments highlight that `CausGNN` consistently boosts the causal reasoning abilities of established GNNs (e.g., GCN, GrapphSAGE and GAT), yielding superior results not only on standard tasks such as node classification and link prediction, but also in the challenging regimes of noisy and OOD scenarios. Our key contributions are outlined below:

- **A Causal Perspective on GNN Limitations.** We theoretically demonstrate that GNNs based on the neighbor aggregation paradigm are fundamentally limited to learning correlational mappings $P(Y|X)$ (please see the Section 3).

- **A Novel Framework.** We propose the Principle of Causal Alignment and provide a model instance `CausGNN`, a general framework that empowers any existing GNN with causal reasoning capabilities.

- **Extensive Empirical Validation.** Extensive experiments validate that our framework effectively enhances a wide range of GNNs. This enhancement is demonstrated by consistently superior performance on node classification and link prediction tasks, and more critically, by substantially improved robustness and generalization under challenging noisy and OOD scenarios.

## 2 PROBLEM FORMULATION

### 2.1 PRELIMINARIES ON GRAPH NEURAL NETWORKS

Graph Neural Networks (GNNs) (Gori et al., 2005; Scarselli et al., 2008) are a class of deep learning models designed to learn a function $\mathcal{F}$ that maps the graph structure and node features to low-dimensional node representations, $\mathbf{H} \in \mathbb{R}^{N \times D}$. Most GNNs follow a message-passing scheme,

where each node iteratively aggregates information from its local neighborhood to update its own representation.

## 2.2 PRELIMINARIES ON CAUSAL INFERENCE

A central goal of causal inference (Pearl, 2009; Bühlmann, 2020) is to distinguish causality from mere statistical correlation. While standard probability theory describes the observational distribution, $P(Y|X)$, which represents the probability of $Y$ given that we have observed $X$, causal analysis seeks to determine the interventional distribution, $P(Y|\text{do}(X))$.

The *do-calculus*, i.e., $\text{do}(X = x)$, denotes a hypothetical intervention where a variable $X$ is forced to take a specific value $x$. This operation simulates an ideal randomized experiment by conceptually severing all causal links that point into $X$, thereby isolating the causal effect of $X$ on other variables from confounding influences.

Pearl (Pearl, 2009) provides a formal framework for computing such interventional distributions from observational data. A key tool within this framework is the backdoor adjustment formula 1. If a set of variables $Z$ satisfies the backdoor criterion(Rumelhart et al., 1986) (i.e., $Z$ blocks all backdoor paths between $X$ and $Y$), the causal effect of $X$ on $Y$ can be calculated from observational probabilities as follows:

$$P(Y|\text{do}(X = x)) = \sum_{z \in \mathcal{Z}} P(Y|X = x, Z = z)P(Z = z). \tag{1}$$

## 2.3 A CAUSAL VIEW ON GNN LEARNING

Let us take a causal look at the GNN modeling and construct a Structural Causal Model (SCM) (Pearl et al., 2016) in Figure 1. We argue that the node's features $X$ and its label $Y$ are jointly influenced by an unobserved, latent variable $Z$, which we term as the environment. $Z$ acts as a common cause, creating a confounding backdoor path. The causal relationships in this SCM are defined as follows:

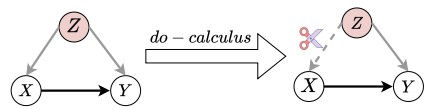
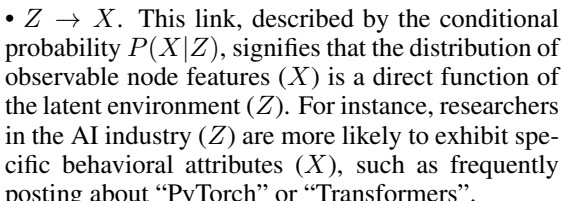

Figure 1: The illustrating causal graph that describes the dependencies among node features $X$, node labels $Y$, and the unobserved environment $Z$. The latent environment $Z$ acts as a confounder, creating a backdoor path $X \leftarrow Z \rightarrow Y$, which can be blocked by the *do-calculus* operator.

• $Z \rightarrow X$. This link, described by the conditional probability $P(X|Z)$, signifies that the distribution of observable node features ($X$) is a direct function of the latent environment ($Z$). For instance, researchers in the AI industry ($Z$) are more likely to exhibit specific behavioral attributes ($X$), such as frequently posting about "PyTorch" or "Transformers".

• $Z \rightarrow Y$. This path represents the impact of the environment on the label. For instance, if a user's social network predominantly consists of individuals in the AI industry ($Z$), its likelihood of being labeled "AI scientist" ($Y$) will increase.

• $X \rightarrow Y$. This is the genuine causal pathway that we aim to isolate and learn. It represents the stable, invariant mechanism where a node's intrinsic properties ($X$) cause its label ($Y$).

A standard GNN model, trained by minimizing an empirical risk over observational data, learns a mapping that approximates the observational conditional probability $P(Y|X)$. Due to the existence of the backdoor path $X \leftarrow Z \rightarrow Y$, the learned probability is a confounded mixture of the true causal effect from $X$ and the spurious correlation induced by $Z$. We exploit the *do-calculus* on the variable $Z$ to remove the backdoor path by estimating $P(Y|do(X))$, as shown in Figure 1.

## 2.4 TASK DEFINITION

To overcome the aforementioned limitation, we aim to develop a general framework that can empower any standard GNN model, denoted by a function $\mathcal{F}_\theta$ with parameters $\theta$, to learn invariant representations. Given a graph $\mathcal{G}$, node features $\mathbf{X}$, and corresponding labels $\mathbf{Y}$ drawn from the ob-

servational distribution $P(\mathbf{X}, \mathbf{Y})$, the goal is to learn an optimal set of parameters, $\theta^*$, by minimizing the empirical risk over the training data.

However, simply minimizing the empirical risk often leads to models that exploit spurious correlations induced by shortcut features. Our task, therefore, is not merely to fit the data, but to do so in a way that reveals the underlying causal structure. This can be formalized as solving a regularized optimization problem, where we supplement the standard supervised objective with a term that enforces a causal inductive bias. We express this objective formally as:

$$\theta^* = \arg\min_{\theta} \quad \mathbb{E}_{(\mathbf{X}, \mathbf{Y}) \sim P(\mathbf{X}, \mathbf{Y})}[\mathcal{L}_{\text{sup}}(\mathcal{F}_{\theta}(\mathbf{X}), \mathbf{Y})] + \mathcal{R}_{\text{causal}}(\mathcal{F}_{\theta}), \tag{2}$$

where $\mathcal{L}_{\text{sup}}$ is the standard supervised loss, which ensures the model fits the observed data. The crucial component is $\mathcal{R}_{\text{causal}}(\mathcal{F}_{\theta})$, a conceptual causal regularizer that forces the model to focus on causally invariant patterns.

## 3   THE LIMITATION OF GNNS

Our argument is that standard GNNs (Kipf, 2016; Hamilton et al., 2017; Veličković et al., 2017; Brody et al., 2021) are limited to learning the observational distribution $P(Y|X)$, which is not limited to a single architecture but applies to the entire neighborhood aggregation paradigm. The core of this paradigm is to update a node's representation by aggregating messages from its neighbors, typically via a weighted summation. We can express this universal aggregation step for a node $v_i$ as:

$$\mathbf{h}'_i = \sigma \left( \sum_{j \in \mathcal{N}_i \cup \{i\}} w_{ij} \cdot \mathbf{W} \mathbf{h}_j \right), \tag{3}$$

where $\mathbf{h}'_i$ is the updated representation, and $w_{ij}$ is the aggregation weight. The key insight is that different GNNs are simply different instantiations of this weighting scheme $w_{ij}$:

- For **GCN**, $w_{ij}$ is a static, pre-defined normalization constant based on node degrees: $w_{ij} = 1/\sqrt{\deg(i)\deg(j)}$.
- For **GraphSAGE** (with mean aggregation), $w_{ij}$ is a uniform average: $w_{ij} = 1/|\mathcal{N}_i|$.
- For **GAT and GATv2**, $w_{ij}$ is a dynamic, learnable attention coefficient $\alpha_{ij}$. Both architectures learn a function $f_{\text{attn}}(\mathbf{h}_i, \mathbf{h}_j)$ to score neighbor importance, differing only in the function's implementation to enhance expressiveness. This score is then normalized via softmax to produce the final weight:

$$w_{ij} = \alpha_{ij} = \text{softmax}_j(f_{\text{attn}}(\mathbf{h}_i, \mathbf{h}_j)). \tag{4}$$

.

From a probabilistic perspective, this universal weighting scheme $w_{ij}$ serves as the model's mechanism for approximating the conditional probability $P(\mathcal{N}_i|X_i)$—that is, which neighborhood context is important given the central node. The subsequent layers (the predictor) then model the distribution $P(Y_i|X_i, \mathcal{N}_i)$. In end-to-end training, the model must learn a single function that maps $X$ to $Y$. This process forces the model to marginalize out the specific neighborhood context, which mathematically collapses the learning objective to the observational probability $P(Y|X)$. The full derivation is as follows:

$$\sum_{\mathcal{N}_i} P(Y_i|X_i, \mathcal{N}_i) P(\mathcal{N}_i|X_i) = \sum_{\mathcal{N}_i} \frac{P(Y_i, X_i, \mathcal{N}_i)}{P(X_i, \mathcal{N}_i)} \cdot \frac{P(X_i, \mathcal{N}_i)}{P(X_i)}$$

$$= \frac{1}{P(X_i)} \sum_{\mathcal{N}_i} P(Y_i, X_i, \mathcal{N}_i) \tag{5}$$

$$= \frac{P(Y_i, X_i)}{P(X_i)} = P(Y_i|X_i).$$

However, the neighborhood context $\mathcal{N}_i$ is not always benign. It sometimes serves as a proxy for the unobserved environmental confounder $Z$. The marginalization process compels the model to absorb

and internalize all observed statistical relationships between neighborhoods and labels. If a spurious correlation exists, it will be incorporated into the distribution $P(Y|X)$.

Consequently, the learned $P(Y|X)$ is a confounded mixture of the true causal effect ($X \rightarrow Y$) and the spurious correlation induced by the latent confounder $Z$ through the backdoor path ($X \leftarrow Z \rightarrow Y$). The model is unable to distinguish the genuine signal from the shortcut features, rendering it vulnerable to spurious patterns and leading to poor generalization.

## 4 METHODOLOGY

To address the challenge of learning causal relationships from confounded graph data, we present our solution in a structured manner. We begin by formally introducing the Principle of Causal Alignment, which serves as the theoretical foundation of our work. Subsequently, we present `CausGNN`, a concrete framework that operationalizes this principle within an efficient teacher-student architecture.

### 4.1 THE PRINCIPLE OF CAUSAL ALIGNMENT

Our core idea is to guide a standard GNN to learn invariant causal relationships by forcing its predictions to align with a constructed, causally-debiased distribution. We formalize this idea as follows:

**Definition 4.1.** *(Principle of Causal Alignment) A GNN model $F_\theta$ satisfies the Principle of Causal Alignment if it satisfies the following two criteria:*

1. *minimizes the empirical risk on observational data;*

2. *aligns its predictive output with an ideal, causally-debiased distribution $P^*(Y|X)$.*

Guided by this principle, we design the learning strategy as a joint optimization problem over the target GNN's parameters $\theta$ and the parameters $\psi$ of the interventional distribution approximator:

$$\min_{\theta, \psi} \mathcal{L} = \mathbb{E}_{(X,Y)}[\mathcal{L}_{\text{sup}}(F_\theta(X), Y)] + \lambda \cdot \mathbb{E}_X[D_{KL}([P_\psi(Y|X)] \parallel P_\theta(Y|X))], \quad (6)$$

where the ideal distribution $P^*(Y|X)$ is approximated by a learnable distribution $P_\psi(Y|X)$, which is parameterized by $\psi$. Inspired by causal inference (Pearl, 2009; Bühlmann, 2020), in practice, we obtain $P_\psi(Y|X)$ by learning to compute the interventional distribution $P(Y|\text{do}(X))$ via Equation 1. $\mathcal{L}_{\text{sup}}$ is the standard supervised loss (e.g., cross-entropy) for $F_\theta$, $D_{KL}(\cdot||\cdot)$ is the Kullback-Leibler divergence (Kullback & Leibler, 1951) and $\lambda$ is a hyperparameter balancing the two objectives. The second term, which we term the Causal Alignment Regularizer, forces $P_\theta(Y|X)$ to be aligned with the learnable causal target $P_\psi(Y|X)$.

**Justification.** The capacity of our principle to discover invariant relationships is theoretically justified in Appendix B. The intuition is that if our constructed target distribution $P^*(Y|X)$ is inherently invariant to environmental shifts, then by aligning $P_\theta(Y|X)$ and this stable target, our objective function effectively discourages reliance on shortcut features and thereby incentivizes the learning of causal representations.

### 4.2 MODEL INSTANCE

In this section, we give a model instance based on the Principle of Causal Alignment, denoted as `CausGNN`. The overall framework is shown in Figure 2.

#### 4.2.1 TEACHER: CONSTRUCTING THE CAUSAL TARGET VIA INTERVENTION

The first step in implementing the Principle of Causal Alignment is to construct the learnable causal distribution $P_\psi(Y|X)$, which we regard as the output of the teacher model. The teacher model is designed to generate environmentally invariant representations. This is achieved by learning to compute the interventional distribution $P(Y|\text{do}(X))$, which involves three main actions: first, explicitly modeling and marginalizing the influence of the latent confounder $Z$; then constructing intervention pairs to simulate diverse environments.

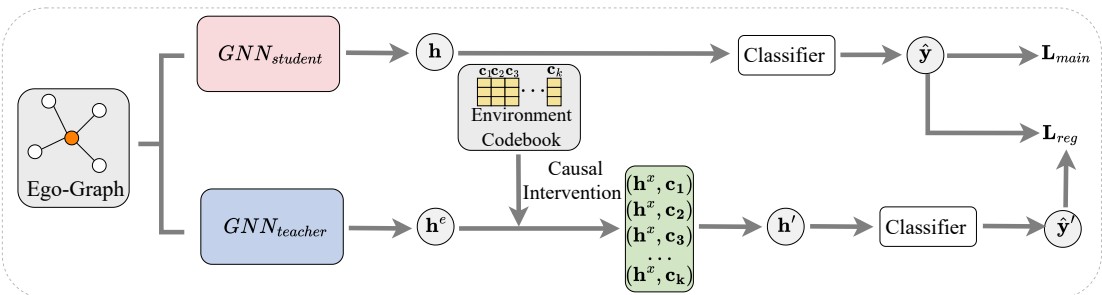

Figure 2: An overview of the `CausGNN` framework, which consists of a teacher GNN and a student GNN. The teacher provides a causally-debiased signal by approximating the interventional distribution, while the student learns to mimic this causal logic. During inference, only the student GNN is employed.

**Modeling Confounders** ($Z$). Since the confounder $Z$ is unobserved, we propose to model its discrete states using a learnable environment codebook, denoted by $\mathbf{C} = \{\mathbf{c}_1, \ldots, \mathbf{c}_K\} \subset \mathbb{R}^{D_e}$. Each vector $\mathbf{c}_k$ serves as a prototype representing one of the $K$ latent environments, and $D_e$ is the dimension of these prototypes. For each node $v_i$, we first extract an environment-aware representation $\mathbf{h}_i^e$ from its initial features $\mathbf{x}_i$ (e.g., via a GNN).

**Environment Stratification and Prior Estimation** ($P(Z)$). To eliminate the influence of confounder $Z$ as much as possible, we need to account for all its stratifications. We compute the probability that a node $v_i$ belongs to environment $k$ using a soft assignment mechanism based on the distance between its representation $\mathbf{h}_i^e$ and each prototype $\mathbf{c}_k$:

$$q_{ik} = P(Z = k|v_i) = \frac{\exp(-\|\mathbf{h}_i^e - \mathbf{c}_k\|_2^2/\tau)}{\sum_{j=1}^{K} \exp(-\|\mathbf{h}_i^e - \mathbf{c}_j\|_2^2/\tau)}, \tag{7}$$

where $\tau$ is a temperature hyperparameter. Next, we estimate the global prior probability of each environment, $P(Z = k)$, by averaging the assignment probabilities $q_{ik}$ over all nodes in a training batch $\mathcal{B}$.

$$p_k = P(Z = k) \approx \mathbb{E}_{j \in \mathcal{B}}[q_{jk}] = \frac{1}{|\mathcal{B}|} \sum_{j \in \mathcal{B}} q_{jk}. \tag{8}$$

**Approximating the Interventional Prediction** ($P(Y|do(X))$). With the environment priors $p_k$ estimated, we can now construct the final intervened representation for each node $v_i$. For each of the $K$ possible environments, we will perform the *do-calculus*. Note that $\mathbf{h}_i^e$ is an environment-aware representation, designed to capture contextual signals for environment matching. In contrast, we now require an environment-agnostic representation, denoted as $\mathbf{h}_i^x$, to capture the node's intrinsic features. This could be the raw features $\mathbf{x}_i$, or features passed through a simple MLP projection to avoid neighborhood contamination. Specifically, we simulate conditioning on $Z = k$ by concatenating the node's intrinsic causal features $\mathbf{h}_i^x$ with the corresponding environment prototype $\mathbf{c}_k$. This pair is then fed into a classifier to produce an environment-specific prediction logit $\mathbf{m}_{ik}$.

$$\mathbf{m}_{ik} = \text{MLP}_{\text{msg}}(\text{concat}(\mathbf{h}_i^x, \mathbf{c}_k)). \tag{9}$$

This step models the conditional term $P(Y|X, Z = k)$ in Equation 1. Finally, we compute the causally-intervened representation $\mathbf{h}_i'$ by taking the weighted average of these environment-specific logits, using $p_k$ as weights. This process can be formally expressed as:

$$\mathbf{h}_i' = \sum_{k=1}^{K} p_k \cdot \mathbf{m}_{ik} = \sum_{k=1}^{K} P(Z = k) \cdot \text{MLP}_{\text{msg}}(\text{concat}(\mathbf{h}_i^x, \mathbf{c}_k)). \tag{10}$$

The resulting representation $\mathbf{h}_i'$ is, by construction, debiased, as the influence of the confounder $Z$ has been explicitly marginalized out. The final teacher's predictive distribution, $P_\psi(Y|X_i)$, is then obtained by applying a softmax function to $\mathbf{h}_i'$. The parameters of the teacher module (the codebook $\mathbf{C}$, the encoders, and the classifier) are collectively denoted by $\psi$.

### 4.2.2 STUDENT: LEARNING INVARIANCE VIA CAUSAL ALIGNMENT

The "student" is any standard GNN model, $F_\theta$, that we aim to empower. It processes the graph $\mathcal{G}$ and features $\mathbf{X}$ as usual to compute node embeddings $\mathbf{h}_i = F_\theta(\mathbf{A}, \mathbf{X})_i$. These embeddings are then fed into a classifier to produce the student's own predictions, $\hat{\mathbf{y}}_i$, and its predictive distribution, $P_\theta(Y|X_i) = \text{Softmax}(\hat{\mathbf{y}}_i)$.

### 4.2.3 OVERALL LEARNING OBJECTIVE

The `CausGNN` framework is trained end-to-end by optimizing the joint objective $\mathcal{L}$ as defined in Equation 6. At inference time, the teacher module is discarded, and predictions are made solely using the trained student GNN, $F_\theta$, incurring no additional computational cost.

## 5 EXPERIMENT

In this section, we conduct extensive experiments to validate the effectiveness and versatility of our proposed `CausGNN` framework. We aim to answer the following research questions: **(RQ1)** Can `CausGNN` consistently improve the performance of various standard GNNs on fundamental graph learning tasks? **(RQ2)** Does our framework enhance the model's robustness against out-of-distribution challenges and structural noise?

### 5.1 EXPERIMENTAL SETUP

Our empirical validation is conducted on a diverse suite of large-scale OGB benchmark datasets (Hu et al., 2020), including *ogbn-arxiv*, *ogbn-products*, *ogbn-mag*, *ogbn-proteins* for node classification, *ogbl-collab*, *ogbl-citation2* for link prediction, and specialized OOD benchmarks (Wu et al., 2024) such as *arxiv-ood* and *twitch-ood*. We evaluate the performance of our framework by applying it to several widely-used GNN architectures—GCN (Kipf, 2016), GraphSAGE (Hamilton et al., 2017), GAT (Veličković et al., 2017), and GATv2 (Brody et al., 2021)—and comparing the enhanced models against their original versions. We report the mean and standard deviation over multiple runs for all experiments to ensure reliable conclusions. The primary evaluation metric is accuracy for classification tasks and Hits@K for link prediction. Full details regarding dataset statistics, baselines, and hyperparameter settings can be found in Appendix C.

To answer our research questions, we structure our experiments into three main parts. First, we evaluate the performance of `CausGNN` on standard, in-distribution benchmarks for node classification and link prediction (**RQ1**). Second, we assess the framework's generalization capability on specialized OOD datasets (**RQ2**). Third, we conduct a robustness analysis by introducing varying levels of structural noise to the training data (**RQ2**).

### 5.2 MAIN RESULTS ON STANDARD BENCHMARKS (RQ1)

**Node Classification.** Table 1 presents the node classification accuracy on four large-scale OGB datasets, where the results of the baselines are taken from GATv2 (Brody et al., 2021). The results provide strong evidence for the effectiveness of our `CausGNN` framework. A key observation is the universality of performance enhancement: across all four diverse datasets and all four baseline GNN architectures, the "`CausGNN(*)`" variant significantly outperforms its original counterpart.

This improvement is particularly pronounced on complex, heterogeneous graphs like ogbn-mag, where `CausGNN`(GATv2) improves upon the baseline by over $1.3\%$. We attribute this to the fact that such graphs often contain more subtle and varied community structures, which act as powerful confounders. Standard GNNs are prone to latching onto these spurious structural correlations. By compelling the model to align with a causally-debiased teacher, `CausGNN` effectively regularizes the learning process, steering the model away from these shortcut features and towards more generalizable, content-based features. Even on a relatively homogeneous graph like *ogbn-arxiv*, the consistent gains (e.g., GCN improves from $71.74\%$ to $72.42\%$) suggest that latent confounders are pervasive and that our method successfully mitigates their negative impact, leading to a more robust predictive model even in standard in-distribution settings.

Table 1: Node classification accuracy (%) on large-scale OGB datasets. Mean and standard deviation over multiple runs are reported. Results for models enhanced by our framework are highlighted in gray.

| Model | Attn. Heads | ogbn-arxiv | ogbn-products | ogbn-mag | ogbn-proteins |
|---|---|---|---|---|---|
| GCN | 0 | $71.74_{\pm 0.29}$ | $78.97_{\pm 0.33}$ | $30.43_{\pm 0.25}$ | $72.51_{\pm 0.35}$ |
| CausGNN(GCN) | 0 | $72.42_{\pm 0.32}$ | $79.73_{\pm 0.18}$ | $31.97_{\pm 0.39}$ | $72.91_{\pm 0.35}$ |
| GraphSAGE | 0 | $71.49_{\pm 0.27}$ | $78.70_{\pm 0.36}$ | $31.53_{\pm 0.15}$ | $77.68_{\pm 0.20}$ |
| CausGNN(GraphSAGE) | 0 | $72.19_{\pm 0.49}$ | $79.52_{\pm 0.09}$ | $32.29_{\pm 0.12}$ | $78.44_{\pm 0.11}$ |
| GAT | 1 | $71.59_{\pm 0.38}$ | $79.04_{\pm 0.54}$ | $32.20_{\pm 1.46}$ | $70.77_{\pm 5.79}$ |
|  | 8 | $71.54_{\pm 0.30}$ | $77.23_{\pm 2.37}$ | $31.75_{\pm 1.60}$ | $78.63_{\pm 1.62}$ |
| CausGNN(GAT) | 1 | $72.83_{\pm 0.17}$ | $80.44_{\pm 0.16}$ | $32.93_{\pm 1.22}$ | $72.02_{\pm 3.37}$ |
|  | 8 | $73.06_{\pm 0.22}$ | $80.47_{\pm 0.37}$ | $32.85_{\pm 0.84}$ | $79.27_{\pm 3.52}$ |
| GATv2 | 1 | $71.78_{\pm 0.18}$ | $80.63_{\pm 0.70}$ | $32.61_{\pm 0.44}$ | $77.23_{\pm 3.32}$ |
|  | 8 | $71.87_{\pm 0.25}$ | $78.46_{\pm 2.45}$ | $32.52_{\pm 0.39}$ | $79.52_{\pm 0.55}$ |
| CausGNN(GATv2) | 1 | $73.15_{\pm 0.35}$ | $\mathbf{81.72}_{\pm 0.33}$ | $33.94_{\pm 0.44}$ | $78.47_{\pm 1.63}$ |
|  | 8 | $\mathbf{73.30}_{\pm 0.18}$ | $81.53_{\pm 0.31}$ | $\mathbf{34.02}_{\pm 0.31}$ | $\mathbf{80.17}_{\pm 2.37}$ |

**Link Prediction.** To verify the applicability of our framework to edge-level reasoning, we evaluate it on two link prediction benchmarks. The results, shown in Table 4, demonstrate that the benefits of causal regularization extend to this domain. For instance, on the large-scale *ogbl-citation2* network, CausGNN(GATv2) improves the link prediction score by nearly one absolute point. This is significant because link prediction relies heavily on understanding the underlying graph structure. The improvement suggests that a standard GNN might overfit to incidental or transitive connections common within a specific research community (a confounder). Our framework encourages the model to learn a more fundamental notion of relational causality—what truly makes two entities likely to connect—rather than simply memorizing common structural patterns. This results in a more accurate model for predicting unobserved graph topology.

## 5.3 Generalization and Robustness Analysis (RQ2)

**Out-of-Distribution (OOD) Generalization.** The core hypothesis of our work is that learning causal representations directly translates to superior generalization under distribution shifts. We test this on the *arxiv-ood* and *twitch-ood* benchmarks. The results, presented in Table 2, offer compelling validation for this claim. The performance gap between the "CausGNN(*)" variants and their baselines is markedly wider in this OOD setting compared to the in-distribution tasks.

Notably, on the *twitch-ood* benchmark, which simulates temporal shifts in user behavior and community structure, CausGNN(GATv2) outperforms the vanilla GATv2 by a significant margin of over 1.4% on the OOD-2 split. This is a crucial finding: standard GATv2, despite its dynamic attention, learns patterns specific to the training "environment" (e.g., popular trends in the training period). When this environment changes in the test set, these learned patterns become invalid. In contrast, CausGNN's teacher branch, through backdoor adjustment, provides a predictive signal that is invariant to these environmental changes. By distilling this invariant logic, the student model learns to rely on features that are causally linked to the label, ensuring its performance remains stable even when the background context shifts. This demonstrates the practical value of learning to approximate $P(Y|do(X))$.

**Robustness to Structural Noise.** To empirically validate our framework's ability to learn invariant representations, we conducted a robustness analysis against structural noise. This experiment serves a dual purpose: it directly tests the model's resilience to perturbations and, more importantly, probes its capacity to mitigate spurious correlations introduced by the graph structure, which acts as a proxy for the confounding environment $Z$. We injecte structural noise by randomly adding non-existing edges to the graph, varying the noise ratio $p$ from 0.0 to 0.5. The results are presented in Figure 3. Please see the Appendix D.2 for detailed analysis.

Table 2: Out-of-distribution generalization accuracy (%) on the *arxiv-ood* and *twitch-ood* datasets.

| Method | arxiv | | | twitch | | |
|---|---|---|---|---|---|---|
| | OOD 1 | OOD 2 | OOD 3 | OOD 1 | OOD 2 | OOD 3 |
| GCN | 54.76 $\pm$0.23 | 52.05 $\pm$0.14 | 44.57 $\pm$0.38 | 64.67 $\pm$0.22 | 51.07 $\pm$0.36 | 61.72 $\pm$0.12 |
| `CausGNN`(GCN ) | 56.92 $\pm$0.24 | 54.25 $\pm$0.49 | 46.66 $\pm$0.83 | 67.95 $\pm$0.27 | 53.53 $\pm$0.02 | 63.91 $\pm$0.08 |
| GraphSAGE | 55.04 $\pm$0.24 | 52.18 $\pm$0.26 | 44.45 $\pm$0.14 | 64.76 $\pm$0.25 | 51.25 $\pm$0.17 | 61.86 $\pm$0.12 |
| `CausGNN`(GraphSAGE) | 57.35 $\pm$0.11 | 54.81 $\pm$0.33 | 46.89 $\pm$0.37 | 67.15 $\pm$0.13 | 53.66 $\pm$0.02 | 63.80 $\pm$0.06 |
| GAT | 55.13 $\pm$0.15 | 51.94 $\pm$0.33 | 44.27 $\pm$0.14 | 65.38 $\pm$0.71 | 51.14 $\pm$0.17 | 61.52 $\pm$0.25 |
| `CausGNN`(GAT) | 57.67 $\pm$0.46 | 54.52 $\pm$0.51 | 46.09 $\pm$0.58 | **67.76** $\pm$0.30 | 53.78 $\pm$0.04 | 63.65 $\pm$0.08 |
| GATv2 | 55.20 $\pm$0.74 | 52.94 $\pm$1.03 | 44.83 $\pm$0.49 | 64.30 $\pm$0.23 | 51.01 $\pm$0.14 | 61.22 $\pm$0.47 |
| `CausGNN`(GATv2) | **58.10** $\pm$1.22 | **54.92** $\pm$1.93 | **47.18** $\pm$0.62 | 67.62 $\pm$0.74 | **53.88** $\pm$0.07 | **63.86** $\pm$0.02 |

## 6 RELATED WORKS

**Graph Neural Networks.** The dominant paradigm for GNNs is neighborhood aggregation. Its evolution has produced a rich family of architectures, from the foundational spectral-based Graph Convolutional Network (GCN (Kipf, 2016)), to inductive methods like GraphSAGE (Hamilton et al., 2017), and attention-based models such as GAT (Veličković et al., 2017) and its more expressive successor, GATv2 (Brody et al., 2021). Further work has explored theoretical expressive limits with models like GIN (Xu et al., 2018) and developed hierarchical pooling mechanisms for graph-level tasks (Ying et al., 2018). Our work is orthogonal to architectural design; `CausGNN` is a general training framework that can empower any of these GNNs with causal invariance.

**Causal Inference on Graphs.** Recent efforts to instill causality in GNNs largely focus on improving OOD generalization for graph-level tasks. A major direction is to disentangle the input graph into causal and spurious components, often through rationale discovery or attention mechanisms (Sui et al., 2022; Wu et al., 2022; Fan et al., 2022; Chen et al., 2022). Other approaches either integrate domain-specific models for particular tasks (Wang et al., 2022) or regularize the final output representations to mitigate confounding (Gao et al., 2024). However, they often introduce complex architectural designs and are limited to graph-level OOD generalization. In contrast, our framework is a lightweight, general solution to node-level tasks across in-distribution, OOD, and noisy settings, requiring no architectural changes and thus not sacrificing inference efficiency. Furthermore, while prior work (Wu et al., 2022; Sui et al., 2022; Wu et al., 2024) acknowledges shortcut features, our contribution is foundational: we are the first to trace their origin to the neighborhood aggregation mechanism and formally prove that this paradigm inherently limits GNNs to learning observational correlations.

## 7 CONCLUSION

In this paper, we address a fundamental limitation of existing GNNs: their inherent tendency to learn spurious correlations rather than true causal relationships. We argue that this vulnerability, which stems from their probabilistic nature of modeling $P(Y|X)$, leads to poor generalization and a lack of robustness. To overcome this, we introduce the Principle of Causal Alignment and propose `CausGNN`, a general framework that empowers any standard GNN with the ability to learn causally-invariant representations. By operationalizing the backdoor adjustment criterion, our framework guides a student GNN to align with a causally-debiased teacher, effectively compelling it to learn robust, environment-insensitive features without requiring any architectural modifications. Extensive experiments on a wide range of benchmarks demonstrated that `CausGNN` consistently enhances the performance of various GNNs on standard node classification and link prediction tasks. More importantly, our framework yields significant improvements in challenging out-of-distribution generalization and noise scenarios.

# 8 REPRODUCIBILITY STATEMENT

To ensure the reproducibility of our work, we provide a code implementation of our method via an anonymous link `https://anonymous.4open.science/r/CausGNN/`. All datasets used in our experiments are publicly available from OGB (Hu et al., 2020). Furthermore, the theoretical proofs for the effectiveness of our proposed method are included in Appendix B.

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

## A  LLM Disclosure Statement

All the content of this paper, including data collection, code implementation, and writing, did not use any generative AI tools.

## B  Theoretical Justification for the Principle of Causal Alignment

In this section, we provide a formal theoretical justification for our proposed Principle of Causal Alignment. We aim to prove that a model trained under our objective is guaranteed to learn representations that are invariant to environmental confounders.

### B.1  Formal Setup

Let us first formalize the learning setup. The data is generated from an underlying Structural Causal Model (SCM) with the joint distribution $P(X, Y, Z) = P(Z)P(X|Z)P(Y|X, Z)$, where $Z$ is the latent environmental confounder.

We define the environment-specific risk for a model $F_\theta$ with parameters $\theta$ as the expected loss within a specific environment $z$:

$$R(\theta|z) \triangleq \mathbb{E}_{(X,Y)\sim P(X,Y|Z=z)}[\mathcal{L}_{\sup}(F_\theta(X), Y)], \tag{11}$$

where $\mathcal{L}_{\sup}$ is the supervised loss. A model is considered environment-invariant if its risk is constant across all environments, i.e., $R(\theta|z_1) = R(\theta|z_2)$ for any $z_1, z_2 \in \mathcal{Z}$, which implies $\mathrm{Var}_{z\sim P(Z)}[R(\theta|z)] = 0$.

Our proposed learning objective is (from Equation 6 in the main paper):

$$\min_{\theta,\psi} \mathcal{L}(\theta, \psi) \triangleq \mathbb{E}_{(X,Y)}[\mathcal{L}_{\sup}(F_\theta(X), Y)] + \lambda \cdot \mathbb{E}_X[D_{KL}(\mathrm{sg}[P_\psi(Y|X)] \parallel P_\theta(Y|X))]. \tag{12}$$

### B.2  Assumptions

Our proof relies on the following standard assumptions:

**Definition B.1.** *(Sufficient Approximator) The model family for the causal approximator (parameterized by $\psi$) is sufficiently expressive, and the optimization is effective, such that at the optimum $\psi^*$, its predictive distribution perfectly models the true interventional distribution: $P_{\psi^*}(Y|X) = P(Y|do(X)) = \sum_{z\in\mathcal{Z}} P(Y|X, Z = z)P(Z = z)$.*

**Definition B.2.** *(Sufficient Main Model) The main GNN model family $\{F_\theta\}$ is sufficiently expressive such that there exists a set of parameters $\theta'$ that can perfectly replicate the optimal approximator's distribution: $\exists \theta'$ such that $P_{\theta'}(Y|X) = P_{\psi^*}(Y|X)$.*

**Definition B.3.** *(Identifiability) The causal relationship $P(Y|X, Z)$ is identifiable from the data. We also assume the KL-divergence and the supervised loss are non-negative and are zero if and only if their arguments are identical.*

### B.3  Proof of Invariance

**Definition B.4.** *Under the stated assumptions, the optimal main model $F_{\theta^*}$ that minimizes the Causal Alignment objective is environment-invariant. That is, $\mathrm{Var}_{z\sim P(Z)}[R(\theta^*|z)] = 0$.*

*Proof.* Let $(\theta^*, \psi^*)$ be the parameters that minimize the objective $\mathcal{L}(\theta, \psi)$. According to the *Sufficient Main Model* assumption, there exists a $\theta'$ such that $P_{\theta'}(Y|X) = P_{\psi^*}(Y|X)$. For this $\theta'$, the KL-divergence term in our objective becomes zero: $D_{KL}(P_{\psi^*}(Y|X) \parallel P_{\theta'}(Y|X)) = 0$. Since the overall objective $\mathcal{L}(\theta, \psi)$ is minimized at $(\theta^*, \psi^*)$ and both loss terms are non-negative, the KL-divergence term at the optimum must also be zero. This implies:

$$D_{KL}(P_{\psi^*}(Y|X) \parallel P_{\theta^*}(Y|X)) = 0 \implies P_{\theta^*}(Y|X) = P_{\psi^*}(Y|X) \quad \forall X. \tag{13}$$

This first step shows that our objective successfully forces the optimal main model's predictive distribution to be identical to the optimal approximator's distribution.

Now, from the *Sufficient Approximator* assumption, we know that $P_{\psi^*}(Y|X) = P(Y|\text{do}(X))$. Combining these results, we get:

$$P_{\theta^*}(Y|X) = P(Y|\text{do}(X)). \tag{14}$$

This means the optimal main model has learned to predict according to the true interventional distribution.

Finally, let's analyze the environment-specific risk of this optimal model, $R(\theta^*|z)$. The loss function $\mathcal{L}_{\text{sup}}$ operates on the model's predictive distribution. So, we have:

$$R(\theta^*|z) = \mathbb{E}_{(X,Y)\sim P(X,Y|Z=z)}[\mathcal{L}_{\text{sup}}(P_{\theta^*}(Y|X), Y)]. \tag{15}$$

Substituting the result from the previous step:

$$R(\theta^*|z) = \mathbb{E}_{(X,Y)\sim P(X,Y|Z=z)}[\mathcal{L}_{\text{sup}}(P(Y|\text{do}(X)), Y)]. \tag{16}$$

Let's expand the expectation over the data distribution $P(X,Y|Z = z) = P(Y|X, Z = z)P(X|Z = z)$:

$$R(\theta^*|z) = \int_{X,Y} \mathcal{L}_{\text{sup}}(P(Y|\text{do}(X)), Y)P(Y|X, Z = z)P(X|Z = z)dY\,dX. \tag{17}$$

The key insight is that the predictor itself, $P(Y|\text{do}(X))$, has already marginalized out the influence of the environment $Z$ by its definition. It is a fixed function of $X$ and does not depend on the specific environment $z$ from which the current data sample $(X, Y)$ is drawn. As the risk $R(\theta^*|z)$ is an expectation over the data distribution within environment $z$, and the predictor being evaluated is invariant to $z$, the resulting expected loss becomes independent of $z$.

Therefore, we have:

$$R(\theta^*|z_1) = R(\theta^*|z_2) \quad \text{for any } z_1, z_2 \in \mathcal{Z}. \tag{18}$$

This directly implies that the variance of the environment-specific risk is zero: $\text{Var}_{z\sim P(Z)}[R(\theta^*|z)] = 0$, which concludes the proof. $\square$

## C EXPERIMENTAL DETAILS

### C.1 DATASETS

The detailed statistics of these datasets are summarized in Table 3.

Table 3: Statistics for the datasets used in our experiments, categorized by task: node classification, link prediction, and out-of-distribution generalization.

| Dataset | # Nodes | # Edges | # Classes |
|---|---|---|---|
| *Node Classification* | | | |
| **ogbn-arxiv** | 169 343 | 1 166 243 | 40 |
| **ogbn-products** | 2 449 029 | 61 859 140 | 47 |
| **ogbn-mag** | 1 939 743 | 21 111 007 | 349 |
| **ogbn-proteins** | 132 534 | 39 561 252 | 112 |
| *Link Prediction* | | | |
| **ogbl-collab** | 235 868 | 1 285 465 | - |
| **ogbl-citation2** | 2 927 963 | 30 561 187 | - |
| *Out-of-Distribution Generalization* | | | |
| **arxiv-ood** | 169 343 | 1 166 243 | 40 |
| **twitch-ood** | 34 120 | 892 346 | 2 |

**Node Classification Datasets.** We utilize four large-scale benchmark datasets from the Open Graph Benchmark (OGB) library (Hu et al., 2020). *ogbn-arxiv* is a citation network where nodes are computer science papers and edges represent citations, challenging models to predict the subject area of each paper. *ogbn-products* is an Amazon co-purchase network where nodes are products and edges indicate that two products are frequently bought together; here, the objective is to predict the product category. *ogbn-mag* is a heterogeneous academic graph containing papers, authors, and institutions, where the goal is to predict the venue for each paper. *ogbn-proteins* is a protein-protein interaction network where nodes are proteins, and the objective is to determine the presence of various protein functions based on their biological interactions.

**Link Prediction Datasets.** We use two OGB datasets for evaluating link prediction capabilities. *ogbl-collab* is a collaboration network of authors, with the goal of predicting missing co-authorship links. *ogbl-citation2* is a large-scale paper citation network, where the objective is to predict missing citation links between papers.

**Out-of-Distribution (OOD) Datasets.** To evaluate generalization capability under distribution shifts, we follow the setting of (Wu et al., 2024) and employ two specialized OOD benchmark datasets. *arxiv-ood* is a variant of the *ogbn-arxiv* dataset where the training, validation, and testing sets are partitioned by publication year. This setup creates a temporal distribution shift, requiring the model to generalize to future, unseen data distributions. *twitch-ood* is a social network of Twitch users. The distribution shift is induced by partitioning users based on their activity levels, simulating changes in community structure and behavior over time.

## C.2 BASELINES

For our baselines, we select a suite of widely used GNN architectures that are the direct targets for our enhancement framework. These include:

- **GCN** (Kipf, 2016). As a foundational GNN model, GCN adapts the convolution operation from images to graph data by simplifying spectral graph theory. In practice, its aggregation mechanism can be viewed as a weighted average of a node's and its neighbors' feature vectors. The aggregation weights are static, pre-defined normalization constants derived directly from the graph structure (i.e., node degrees), making it a powerful but non-adaptive baseline.

- **GraphSAGE** (Hamilton et al., 2017). GraphSAGE represents a significant step towards inductive learning on graphs, allowing models to generalize to unseen nodes. Instead of learning fixed embeddings for each node, it learns aggregator functions (e.g., mean, max-pooling, or an LSTM-based aggregator) that define how to gather information from a node's local neighborhood. This flexible, learnable aggregation makes it a widely used and powerful spatial GNN.

- **GAT** (Veličković et al., 2017). GAT introduced the self-attention mechanism to the graph domain, enabling nodes to assign different importance weights to their neighbors during aggregation. Unlike GCN's fixed weights, GAT's attention coefficients are learnable and dependent on the features of the interacting nodes, which allows the model to focus on more relevant information.

- **GATv2** (Brody et al., 2021). GATv2 is a direct successor to GAT, designed to fix a subtle limitation in the original attention mechanism. It demonstrates that the original GAT's attention function is "static" in its expressiveness, meaning the ranking of neighbor importance is not fully conditioned on the querying node. By modifying the order of operations within the attention computation, GATv2 achieves a more powerful and truly "dynamic" attention mechanism.

## C.3 EXPERIMENTAL CONFIGURATION

All experiments are conducted on a server equipped with NVIDIA A100 GPUs. Our framework and all baseline models are implemented using PyTorch (Paszke et al., 2019) and PyTorch Geometric (PyG) (Fey & Lenssen, 2019).

Table 4: Link prediction performance (Hits@K) on OGB benchmarks.

| Model | Attn. Heads | ogbl-collab | ogbl-citation2 |
|---|---|---|---|
| GCN | 0 | $44.75_{\pm 1.07}$ | $80.04_{\pm 0.25}$ |
| CausGNN(GCN ) | 0 | $45.32_{\pm 0.62}$ | $80.72_{\pm 0.27}$ |
| GraphSAGE | 0 | $48.10_{\pm 0.26}$ | $80.44_{\pm 0.17}$ |
| CausGNN(GraphSAGE) | 0 | $\mathbf{48.38}_{\pm 0.74}$ | $80.93_{\pm 0.14}$ |
| GAT | 1 | $39.32_{\pm 3.26}$ | $79.84_{\pm 0.19}$ |
| | 8 | $42.37_{\pm 2.99}$ | $75.95_{\pm 1.31}$ |
| CausGNN(GAT) | 1 | $40.07_{\pm 0.23}$ | $80.35_{\pm 1.16}$ |
| | 8 | $43.06_{\pm 0.36}$ | $78.47_{\pm 0.47}$ |
| GATv2 | 1 | $42.00_{\pm 2.40}$ | $80.33_{\pm 0.13}$ |
| | 8 | $42.85_{\pm 2.64}$ | $80.14_{\pm 0.71}$ |
| CausGNN(GATv2) | 1 | $43.15_{\pm 1.05}$ | $80.76_{\pm 0.29}$ |
| | 8 | $43.62_{\pm 0.64}$ | $\mathbf{81.03}_{\pm 0.52}$ |

To ensure the reliability and robustness of our results, all experiments are repeated for 10 runs with different random seeds. The mean and standard deviation of the performance metrics across these 10 runs are reported in all result tables.

## C.4 HYPERPARAMETER SETTINGS

The hyperparameter configurations of our experiments are configured as follows:

**Baseline GNNs (GCN, GraphSAGE, GAT, GATv2).** Our approach to baseline hyperparameters varied by experimental setting:

- **For In-Distribution Tasks:** For the standard node classification (Table 1) and link prediction (Table 4) benchmarks, the results for the baselins are adopted directly from the GATv2 paper (Brody et al., 2021) to ensure a fair comparison against established, well-tuned results.

- **For OOD and Noise Robustness Tasks:** For the out-of-distribution (OOD) generalization (Table 2) and noise robustness (Figure3) experiments, we perform a thorough grid search to find the optimal hyperparameters for each baseline. The search space is defined as follows:

  - Learning Rate: Searched within $\{0.01, 0.005, 0.001\}$.
  - Weight Decay: Set to $5 \times 10^{-4}$.
  - Number of Layers: Set to 2.
  - Hidden Dimension: Set to 128 for all layers.
  - Dropout Rate: Tuned within $\{0.3, 0.5, 0.6, 0.8\}$.
  - Attention Heads (for GAT/GATv2): Set to 1 for OOD/noise experiments.

**Our Framework.** For all experiments adopting our CausGNN framework (i.e., across all result tables and figures), we perform a grid search on the following hyperparameters:

- **Causal regularizer weights** ($\lambda$): The weight for the causal regularization term is searched in the set $\{0.0, 0.2, 0.5.0.8, 1.0\}$. We find $\lambda = 0.8$ to be a robust choice for most datasets.

- **Number of Environment Prototypes** ($K$): The size of the environment codebook is selected from $\{3, 5, 8, 10, 15\}$. We find $K = 10$ to be a robust choice for most datasets.

- **Environment Dimension** ($D_e$): The embedding dimension for the environment prototypes is set to 128.

- **Temperature** ($\tau$): The temperature for the soft assignment in environment stratification is fixed at 1.0.

# D  ADDITIONAL EXPERIMENTAL RESULTS

## D.1  LINK PREDICTION

Results for link-prediction are shown in Table 4.

## D.2  ROBUSTNESS TO STRUCTURAL NOISE

On the homogeneous *ogbn-arxiv* dataset (Figure 3a), where the neighborhood is composed of a single relation type (citations), *CausGNN(GAT)* maintains a higher accuracy across all tested noise levels. As the noise ratio $p$ increases from $0.0$ to $0.5$, the baseline GAT's accuracy degrades by $5.5\%$. The proposed CausGNN(GAT) exhibits a smaller degradation of $4.5\%$. This result is consistent with our expectation: by approximating the interventional distribution $P(Y|\mathrm{do}(X))$, the model is incentivized to learn a mapping that depends on the node's intrinsic features rather than on the easily corrupted structural context. The performance gap between the two models widens as the noise ratio increases, which indicates that the representations learned by CausGNN possess a higher degree of invariance to structural perturbations.

The superiority of our framework is further underscored on the more challenging heterogeneous ogbn-mag dataset (Figure 3b). Heterogeneous graphs introduce various structural relationships (e.g., authors writing papers, papers having topics), resulting in more complex sources of confounded factors. Despite this, the performance of CausGNN(GAT) decreased by $3\%$, which is relatively mild compared to the baselines. This demonstrates that our framework remains effective in more complex environments. Collectively, the consistent improvements across both homogeneous and heterogeneous settings validate the generality and effectiveness of our model..

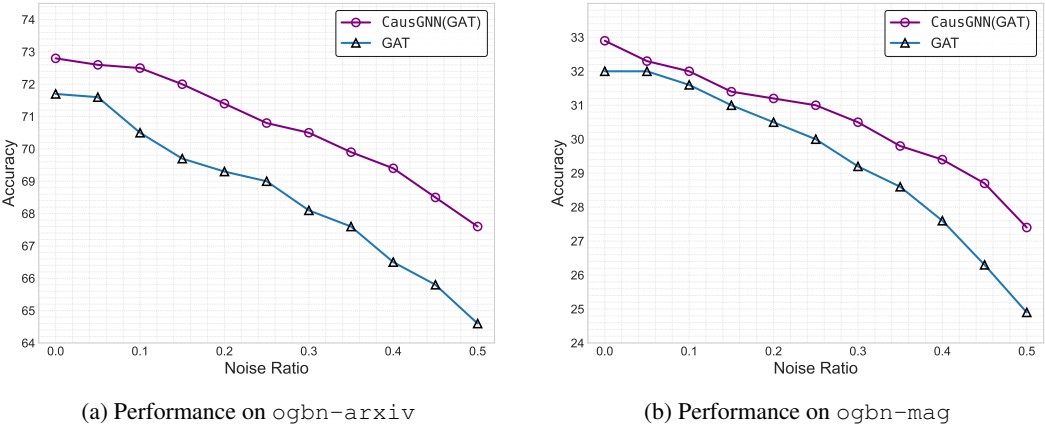

(a) Performance on ogbn-arxiv

(b) Performance on ogbn-mag

Figure 3: Robustness analysis against structural noise. We compare the accuracy of CausGNN(GAT) with its baseline *GAT* on (a) ogbn-arxiv and (b) *ogbn-mag* datasets. The noise ratio on the x-axis represents the proportion of randomly added non-existing edges relative to the original number of edges.

