# OpenReview forum: "A General Framework for Empowering Graph Neural Networks with Causal Invariance"
_ICLR.cc/2026/Conference — ICLR 2026 Conference Withdrawn Submission_

### Official Review · Reviewer_gRMv · 2025-10-30

**Soundness:** 2
**Presentation:** 2
**Contribution:** 2
**Rating:** 2
**Confidence:** 4

**Summary:**

The paper argues that message-passing GNNs inherently learn the observational distribution P(Y|X), making them prone to capturing shortcut correlations from the local neighborhood. To address this, it introduces the Principle of Causal Alignment and a concrete implementation called CausGNN. The framework adopts a teacher–student setup: the teacher network approximates interventional predictions through a backdoor-style marginalization over a latent “environment” variable Z, while the student GNN learns to align its own predictive distribution with the teacher’s output via a KL-divergence regularizer. During inference, only the student model is used. Experiments on multiple OGB benchmarks, including node classification, link prediction, OOD generalization, and noise robustness, are conducted.

**Strengths:**

1. Experiments on a wide range of datasets and ablation study are conducted.

**Weaknesses:**

1. The core claim—that neighborhood aggregation in GNNs inherently leads to confounding and thus poor OOD generalization—is not new. Prior environment-based or invariant graph OOD generalization works (e.g., Invariant Risk Minimization, Causal Attention, Invariant Rationales, Disentangled Substructure GNNs) have already analyzed neighborhood aggregation as a source of environment bias and proposed causal or invariant-based remedies. The contribution of this paper therefore appears to be a re-framing of existing ideas rather than a substantially new conceptual insight.

2. The experimental section compares only with standard GNNs trained under ERM (e.g., GCN, GraphSAGE, GAT, GATv2), which are in fact backbones, not OOD baselines. It omits direct comparisons with prior OOD generalization or invariant learning methods. Without these comparisons, it is difficult to assess how much the proposed method actually advances the state of the art in graph OOD generalization.

3. The reported improvements across OGB benchmarks and OOD settings are mostly within 0.5–1.5% absolute—often within one standard deviation. This suggests that the proposed causal alignment principle provides only limited practical gain over simple ERM training. Statistical significance tests or stronger robustness evaluations would be needed to establish the benefit convincingly.

4. The related-work section is far from comprehensive. Many relevant studies in graph OOD generalization are missing, including but not limited to:

[1] OOD-GNN: Out-of-Distribution Generalization on Graphs via Causal Mechanism Inference

[2] StableGNN: Learning Invariant Representations for Out-of-Distribution Generalization on Graphs

[3] Subgraph Invariant Learning Towards Large-Scale Graph Node Classification

[4] Individual and structural graph information bottlenecks for out-of-distribution generalization

[5] GraphIFE: Rethinking Graph Imbalance Node Classification via Invariant Learning

[6] Enhancing Graph Invariant Learning from a Negative Inference Perspective

[7] Bridging OOD Detection and Generalization: A Graph-Theoretic View

[8] Dissecting the Failure of Invariant Learning on Graphs

[9] Learning invariant representations of graph neural networks via cluster generalization

[10] Joint Learning of Label and Environment Causal Independence for Graph Out-of-Distribution Generalization




These are only a few examples. I strongly encourage the authors to conduct a more comprehensive literature review to better situate the contribution and clarify what new understanding or advantage this work offers over existing causal or invariant frameworks for GNNs.

**Questions:**

N/A

---

### Official Review · Reviewer_k7bC · 2025-10-31

**Soundness:** 3
**Presentation:** 3
**Contribution:** 3
**Rating:** 2
**Confidence:** 4

**Summary:**

This paper discusses the limitations of GNNs, i.e., their reliance on neighborhood aggregation limits them to learning correlations rather than causal relationships, making them susceptible to spurious patterns that affect generalization and robustness. To address this issue, the paper introduces the Principle of Causal Alignment and presents CausGNN, a teacher-student framework. The teacher model learns causal relationships to mitigate the influence of environmental confounders, while the student model is trained to minimize the difference between its predictions and the teacher's output. This framework does not require modifications to the student GNN architecture and imposes no additional computational cost during inference.

**Strengths:**

1. By using a teacher-student framework, all the causal machinery is discarded after training. This means the resulting GNN (the student) has zero additional inference cost compared to its original version, making practical applications easier.

2. The paper provides comprehensive experimental validation for the method.

**Weaknesses:**

1. I believe some of the authors' statements are incorrect, such as the mention of the causal relationship P(Y | X, Z)  in Definition B.3. This P(Y | X, Z)  represents a conditional probability and should not be equated with a causal relationship, which is a fundamental principle in causal theory. Additionally, I do not understand why the authors use "Definition" to refer to all theorems and hypotheses.

2. The method models the (potentially high-dimensional and continuous) unobserved confounder $Z$ using a discrete codebook of $K$ prototypes. This is a strong simplification. While the experiments show it works, the paper does not deeply discuss the limitations of this approximation. If the true environmental confounding is far more complex, this discrete approximation might be insufficient.

**Questions:**

1. The method models the confounder $Z$ with $K$ discrete prototypes48. The appendix mentions $K=10$ is a robust choice49. How sensitive is the model's performance to this hyperparameter? What happens if $K$ is set too low (e.g., $K=2$) or too high? Is there a risk of "model mismatch" if the true confounding is continuous or has a much higher dimensionality?

---

### Official Review · Reviewer_dBDC · 2025-10-31

**Soundness:** 3
**Presentation:** 2
**Contribution:** 2
**Rating:** 4
**Confidence:** 2

**Summary:**

CausGNN not only improves the performance of various classic GNNs but also demonstrates superior robustness to noise. However, its applicability may be limited to classic GNN architectures, and the experimental evaluation remains insufficient.

**Strengths:**

The paper provides a clear background and a clear introduction. It demonstrates performance improvements for classic GNNs.

**Weaknesses:**

1.The motivation of the proposed method originates from the limitations of standard GNNs, while its applicability to more recent GNN architectures remains uncertain.

2.The experimental evaluation is not sufficiently comprehensive. The experiments were conducted only with standard GNNs as backbones, without including more recent architectures as baselines or comparisons with other SOTA methods that enhance GNN performance.

3.The framework appears to be limited to applications involving standard GNNs, with no exploration of its applicability to graph transformers or other advanced GNN architectures. The practical value of the proposed method may be limited.

**Questions:**

1.In the Modeling Confounders section, why can its discrete states be modeled using a learnable environment codebook? Why is the environment codebook considered learnable, and is it actually trainable?

2.The concepts of environment-aware representation and environment-agnostic representation are somewhat abstract. What are their precise definitions? In Figure 2, can h^e be transformed into a combination of h^x$ and the codebook through causal intervention?

---

### Official Review · Reviewer_pGrf · 2025-10-31

**Soundness:** 2
**Presentation:** 2
**Contribution:** 2
**Rating:** 4
**Confidence:** 4

**Summary:**

The paper argues neighborhood-aggregation GNNs learn correlational $P(Y\mid X)$ rather than causal effects, and proposes CausGNN to align models with $P(Y\mid do(X))$. A teacher uses a learnable environment codebook and backdoor-style adjustment to form a causal target distilled to a student via KL, yielding environment-invariant representations and consistent gains on OGB node/link, OOD, and noise settings without inference overhead.

**Strengths:**

- A general training principle (causal alignment) with a clear teacher–student realization; conceptually simple and reusable.
- Teacher performs a backdoor-style approximation with an environment prototype codebook; student aligns via KL. The approach is easy to plug in as a regularizer to many GNNs.
- Promising results across ID/OOD/noise settings on multiple datasets, suggesting potential generality, though broader baseline comparisons are still needed.

**Weaknesses:**

- The core idea is close to some SOTAs (e.g., GDRA: Rationalizing Graph Neural Networks with Data Augmentation), which leverage environment-based invariant learning and distillation/alignment. Please clarify substantive differences and incremental contributions, and compare empirically.
- Lack of direct comparisons with relevant SOTAs. Please include representative, competitive baselines under a unified protocol.
- No systematic analysis/evaluation of environment representation and prototype quality, e.g. quantitative metrics and diagnostic visualizations.
- Limited discussion on identifiability and stability of the learnable environment codebook. Sensitivity to initialization, training drift, prototype collapse/duplication are not thoroughly ablated.

**Questions:**

- Learnable environment: Are the environment prototypes $c_k$ trainable throughout training? Do you freeze them at any stage to avoid target drift? Do random initialization and continuous updates induce distribution shift or instability in the teacher signal?
- Environment representations and prototypes: Can you objectively evaluate the quality of $h^x_i$, $h^e_i$, and $c_k$ (e.g., alignment with community detection/metadata, prototype–class mutual information, node-level environment consistency, qualitative visualizations and failure cases)?
- Cost/benefit: What are the training costs of the teacher branch and the overall trade-offs from distillation?
- Could you cover more SOTAs in related work and experiments, clearly articulate the unique contributions versus these methods, and add representative baselines (e.g., recent causal GNN/IRM/environment-decomposition/distillation methods) to strengthen the empirical conclusions?

---

### Note · Authors · 2025-11-18

I have read and agree with the venue's withdrawal policy on behalf of myself and my co-authors.